# An engaged research study to assess the effect of a 'real-world' dietary intervention on urinary bisphenol A (BPA) levels in teenagers

Tamara S Galloway,[1] Nigel Baglin,[2] Benjamin P Lee,[3] Anna L Kocur,[3] Maggie H Shepherd,[4,5] Anna M Steele,[4,5] BPA Schools Study Consortium,[6, 7, 8, 9, 10, 11] Lorna W Harries[3]

For numbered affiliations see end of article.

**Correspondence to**
Dr Lorna W Harries;
L.W.Harries@exeter.ac.uk

## ABSTRACT

**Objective** Bisphenol A (BPA) has been associated with adverse human health outcomes and exposure to this compound is near-ubiquitous in the Western world. We aimed to examine whether self-moderation of BPA exposure is possible by altering diet in a real-world setting.

**Design** An Engaged Research dietary intervention study designed, implemented and analysed by healthy teenagers from six schools and undertaken in their own homes.

**Participants** A total of 94 students aged between 17 and 19 years from schools in the South West of the UK provided diet diaries and urine samples for analysis.

**Intervention** Researcher participants designed a set of literature-informed guidelines for the reduction of dietary BPA to be followed for 7 days.

**Main outcome measures** Creatinine-adjusted urinary BPA levels were taken before and after the intervention. Information on packaging and food/drink ingested was used to calculate a BPA risk score for anticipated exposure. A qualitative analysis was carried out to identify themes addressing long-term sustainability of the diet.

**Results** BPA was detected in urine of 86% of participants at baseline at a median value of 1.22 ng/mL (IQR 1.99). No effect of the intervention diet on BPA levels was identified overall (P=0.25), but there was a positive association in those participants who showed a drop in urinary BPA concentration postintervention and their initial BPA level (P=0.003). Qualitative analysis identified themes around feelings of lifestyle restriction and the inadequacy of current labelling practices.

**Conclusions** We found no evidence in this self-administered intervention study that it was possible to moderate BPA exposure by diet in a real-world setting. Furthermore, our study participants indicated that they would be unlikely to sustain such a diet long term, due to the difficulty in identifying BPA-free foods.

## Strengths and limitations of this study

► This study represents the largest assessment to date of the potential for moderating one's own bisphenol A (BPA) exposure through diet.
► The study was carried out in a 'real-world' setting rather than a regulated, controlled environment.
► The study was carried out in teenagers, the demographic with among the highest exposure.
► Qualitative analysis reveals challenges with sustaining such a diet.
► Calculation of a risk score is challenging due to the pervasive nature of BPA contamination.

## INTRODUCTION

Bisphenol A (BPA) is one of the world's highest production volume chemicals. It is used in the manufacture of polycarbonate and other plastic consumer products, in heat-resistant papers, dental sealants and in the epoxy resin-based lining of food and drink containers.[1] BPA can be found above the detection limit in the urine of the majority of people worldwide.[2] Concern has been raised for public health, since BPA is classified as an endocrine-disrupting chemical (EDC) which has been linked with several disorders in cell and animal models.[3–5] Several epidemiological studies have also linked outcomes such as type 2 diabetes, cardiovascular disease, obesity and abnormalities of sex hormone levels with BPA levels in human populations[6–10] Epidemiological data in humans has historically been more contentious, however, due to relatively small sample sizes and issues around causality.[11] The Endocrine Society concluded in 2015 that current evidence suggests that BPA and other EDCs may have effects on several reproductive, cardiovascular and metabolic traits in humans.[12] The current opinion of food regulatory bodies such as the European Food Standards Agency (EFSA) is that sufficient uncertainty remains to be able to exclude effects on the reproductive, immune, nervous, metabolic and cardiovascular systems and on cancer development,[3] while the European Chemicals Agency has

recently reclassified BPA as a chemical of very high concern due to its endocrine-disrupting properties.[13]

There has been wide interest in the sources of BPA and the potential for individuals to reduce their own exposure. Human exposure has been reported from inhalation of dust, uptake across the skin from thermal papers and till receipts and release from dental sealants. The main source is the ingestion of food and drink contaminated with BPA leached from packaging materials.[1 14] BPA is rapidly metabolised in the gut wall and liver and removed from the blood by the kidneys, with a terminal half-life of 6 hours after oral ingestion.[15] BPA has been detected in food samples packaged in glass, plastic, paper and paperboard cartons, with an average concentration of 0.46 ng/g, rising to over 700 ng/g for certain canned foods. Conversely, in a dietary intervention study in which 22 volunteers consumed a 3-day fresh food diet which excluded canned or packaged foods, there was a 66% reduction in urinary BPA excretion compared with concentrations before the intervention.[16] This latter study involved full dietary replacement of foodstuffs, an approach which is impractical for the population at large. A follow-up study found that households who followed written recommendations produced by healthcare professionals showed no significant change in their BPA exposure.[17]

We present an alternative, citizen science-based approach, where 108 student volunteers designed and undertook their own intervention diet, following provision of educational materials. We questioned whether adherence to a self-designed and self-administered 'real-world' diet over 7 days would lead to significant reductions in excreted urinary BPA, and if so, whether such a diet was likely to be sustainable in the long term.

## METHODS
### Participant group
We chose adolescents because it has been shown that they have higher concentrations of BPA than adults (aggregated exposure of 1.449 µg/kg body weight per day).[3 18] A total of 108 students aged 17–19 from local schools were initially invited to participate in this engaged research project. Six schools participated in this project (Clyst Vale Community College—14 students; Exeter School—12 students; South Dartmoor Community College—13 students; Honiton Community College—11 students; Exeter College—29 students and Exeter Mathematics School—29 students). Information and samples were available from 94 individuals at both visit 1 and visit 2 and comprise the complete dataset. This represents the largest intervention study in the population demographic with the one of the highest BPA exposures to date.[18] The number of students invited to participate was based on anticipated effect sizes from previous work of this nature,[16] and we allowed for a 10% dropout rate. Students designed all of the materials required for completion of the study (study protocols, food diaries, lifestyle questionnaires, patient information sheets and consent forms (see online supplementary information files 1–6).

### Intervention diet
Students designed a 'real-world' diet designed to reduce consumption of BPA by avoidance of processed foods and foods packaged in known sources of BPA[1 14]; online supplementary information file 1. The study was designed at the University of Exeter as a collaboration between academic staff and participating students and was developed at a series of interactive workshops attended by all parties.

Students were asked to minimise their intake of known sources of BPA according to a set of guidelines that had been codesigned with them based on the known literature. We requested that calorific intake was maintained as near to their usual diet as possible and recorded details of their daily diet including all food and drink, and its associated packaging, in a self-reported food diary (online supplementary information file 2). Adherence was assessed using a 'BPA risk score'; each individual dietary item potentially containing BPA was given a score of 1. Heavily processed items were also scored 1 per item. These scores were collated at the end of the 7-day trial to give a final risk score. An example of scores for a single participant on a single day is given in online supplementary information file 3. Given the short half-life of BPA, we also carried out a secondary analysis considering only the BPA risk score from the 24 hours immediately preceding the second sample. Information on lifestyle factors including sex, body mass index (BMI) and time of urine collection was also collected (online supplementary information file 4). We recognised that there may be a temptation for students to change their diets before the trial based on their new learning. To avoid this, students were also specifically asked not alter their diet before the intervention.

### Sample collection and measurement of urinary BPA
Urine samples were collected into BPA-free bottles (Vacutest Kima, Italy) immediately before and after the intervention, and were frozen at −20°C within 4 hours. Each participant was sampled two times, once at visit 1 before the intervention and once at visit 2 after the intervention. Sample collections were staggered to allow for the large number of participants passing through the facility, but students were sampled during the same time slot at both visits to account for circadian variation in BPA metabolism. The initial samples were collected during the early part of the day just prior to the students commencing the trial. The second samples were taken over the same time period 7 days later, just prior to the students recommencing their usual diet. Samples were transported on dry ice to a commercial laboratory (Rovaltain Research Company, Aixain, France) where analysis of total BPA was assessed by gas chromatography–tandem mass spectrometry. Experimental methods were validated for linearity, detection limit and accuracy and specificity

of quantification based on the Standard NF T 90–201 for determination of xenobiotics. A quality control check of known standards injected every six samples at two levels of concentration (0.5 ng/mL and 5 ng/mL) was quantified with each batch of unknown samples. Water-only samples were included as negative controls. Urinary creatinine was measured at the Royal Devon and Exeter Hospital using the Jaffe method on the Roche P800 platform (Roche, Mannheim, Germany), to allow correction for urine dilution. Results were expressed as a BPA to creatinine ratio. Samples where BPA was detected but quantifying at or around the limits of quantification (LoQ) of 0.1 ng/mL were scored as $LoQ/\sqrt{2}$ according to the method of Hornung and Reed.[19]

## Statistical analysis

The difference between urinary BPA adjusted for creatinine between samples taken at visits 1 and 2 was assessed to generate a ΔBPA continuous variable. BPA risk scores were calculated as a continuous variable. The relationship between urinary BPA levels before and after the 7-day intervention was assessed using a repeated-measure analysis of variance, adjusted for sex, time of sampling and BMI, with and without correction for creatinine. The relationship between urinary BPA at visit 1 and whether or not the participants had lower BPA at visit 2 was also examined by binary logistic regression, adjusted for sex, time of sampling and BMI. Here, samples showing small changes <0.5 ng/mL in either direction were omitted to avoid natural stoichiometric variation around zero. The relationship between change in BPA (ΔBPA) and BPA risk score was assessed by linear regression, adjusted for sex, time of sampling and BMI both with and without adjustment for creatinine. Statistical analysis was carried out using SPSS V.22.

## Impact of following reduced BPA diet on lifestyle

We carried out quantitative and qualitative analysis to address long-term sustainability of the diet. Data on the impact of following the diet on feelings of dietary restriction, time spent sourcing or preparing meals, calorific intake and long-term sustainability were collected via a questionnaire (see online supplementary information file 4). The questionnaire also included a freeform section where participants could write about their experiences following the diet in a non-prescribed fashion for qualitative analysis. Qualitative data were assessed for thematic content by two experienced qualitative researchers. Key themes were independently identified and coded until agreement was reached.

## RESULTS
## Participant characteristics

There were 108 volunteer participants invited to participate in this engaged research study. A small number were absent or unable to produce a urine sample at both visits. A complete dataset was thus received from 94 students.

Information on the characteristics of the study cohort is given in table 1.

BPA was detected in the urine of 86% of subjects at visit 1 prior to the intervention. Missing samples were due to non-attendance of participants or non-provision of a suitable sample. Samples below the LoQ were scored as 0.07 ng/mL $(LoQ/\sqrt{2})$.

## Creatine-adjusted urinary BPA concentrations did not change significantly after following the intervention diet

The median change in creatinine-adjusted urinary BPA between visits (ΔBPA) was 0.05 ng/mL with an IQR of 2.94 ng/mL. We identified no changes in urinary BPA between visits (P=0.25; figure 1A). Three outliers with very high urinary BPA readings at visit 2 were excluded from the analysis, since these samples lay outside the linear range of analysis, so confidence in quantification was poor. No confounding factors included in the analysis were associated with change in BPA (P=0.78, 0.43 and 0.36 for sex, time of sample collection and BMI, respectively). We also identified no change in BPA levels between visits using data uncorrected for creatinine (P=0.20). We also assessed whether participants from different schools showed variable BPA levels at either visit 1 or change in BPA, but no such effects were noted.

Similarly, no relationship between change in urinary BPA (ΔBPA) and BPA risk score was identified (beta coefficient 0.08, SE 0.07, P=0.55; figure 1B). No associations were noted between change in urinary BPA and BPA risk score in data not adjusted for creatinine (P=0.27). We found no association between ΔBPA and BPA risk score when considering only the exposure on the day prior to testing, taking into account the short half-life of BPA (P=0.16 and P=0.33 for adjusted and unadjusted data, respectively).

## Participants with highest starting urinary BPA were more likely to demonstrate lower BPA at visit 2

We found an inverse relationship between initial BPA concentrations and whether a participant had reduced BPA concentrations at visit 2 (P=0.003). These data indicate that the participants in the cohort with the highest creatinine-adjusted urinary BPA concentrations at visit 1 were more likely to demonstrate a drop in their urinary BPA at visit 2 (figure 2).

## Following the intervention diet has significant effects on participant lifestyle

Participants indicated that following the diet had no significant cost implications on family finances, with 50% of participants reporting that it had cost more, and 50% reporting that costs had decreased or remained the same. Although participants did not spend longer preparing their food, 78% of participants reported that their shopping took longer. Calorific intake was not affected for the majority of participants (58%) of participants. A large percentage of the cohort (91%) reported that they felt at least slightly restricted in their food choices and 27% of

**Table 1** Characteristics of the study population

| Unadjusted urinary BPA at visit 1 (n=94) | |
|---|---|
| Median (IQR) | 1.01 (2.01) |
| 95% CI | 1.19 to 2.57 |
| Mean (SD) | 1.88 (2.68) |
| No of samples below LoQ (0.1 ng/mL) | 15 |
| Minimum value (ng/mL) | 0.07 |
| Maximum value (ng/mL) | 13.55 |
| Creatinine-adjusted urinary BPA at visit 1 (n=94) | |
| Median (IQR) | 1.22 (1.99) |
| 95% CI | 1.16 to 1.20 |
| Mean (SD) | 1.58 (1.64) |
| No of samples below LoQ (0.1 ng/mL) | 15 |
| Minimum value (ng/mL) | 0.05 |
| Maximum value (ng/mL) | 8.56 |
| Unadjusted urinary BPA at visit 2 (n=94) | |
| Median (IQR) | 1.47 (2.87) |
| 95% CI | 1.59 to 3.97 |
| Mean (SD) | 2.78 (4.64) |
| No of samples below LoQ (0.1 ng/mL) | 12 |
| Minimum value (ng/mL) | 0.07 |
| Maximum value (ng/mL) | 31.2 |
| Creatinine-adjusted urinary BPA at visit 2 (n=94) | |
| Median (IQR) | 1.24 (2.51) |
| 95% CI | 1.21 to 5.01 |
| Mean (SD) | 3.13 (7.36) |
| No of samples below LoQ (0.1 ng/mL) | 12 |
| Minimum value (ng/mL) | 0.04 |
| Maximum value (ng/mL) | 53.42 |
| Unadjusted ΔBPA (n=94) | |
| Median (IQR) | 0.06 (2.09) |
| 95% CI | 0.05 to 1.75 |
| Mean (SD) | 0.90 (3.32) |
| Minimum value | −6.42 |
| Maximum value | 17.64 |
| Adjusted ΔBPA (n=94) | |
| Median (IQR) | 0.05 (2.94) |
| 95% CI | −0.28 to 3.39 |
| Mean (SD) | 1.55 (7.16) |
| Minimum value | −4.47 |
| Maximum value | 50.38 |
| BPA risk score (n=94) | |
| Median (IQR) | 17.0 (11.0) |
| 95% CI | 15.4 to 18.8 |
| Mean (SD) | 17.1 (6.63) |

Continued

**Table 1** Continued

| Demographics (n=94) | |
|---|---|
| Sex (%) male | 44 |
| Exposure to oestrogens (%) of cohort | 15 |
| BMI, median (IQR) | 20.7 (3.43) |
| BMI, mean (SD) | 21.3 (3.13) |

A complete dataset was available on 94 out of 108 participants. The units of BPA and BMI are ng/mL and kg/m², respectively. Urinary BPA levels are given both as unadjusted data and as a BPA (ng/mL) to creatinine (mg/mL) ratio.
BMI, body mass index; BPA, bisphenol A; LoQ, limit of quantification.

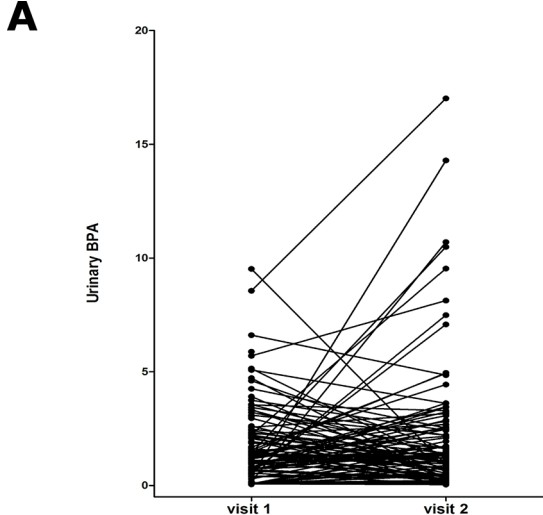

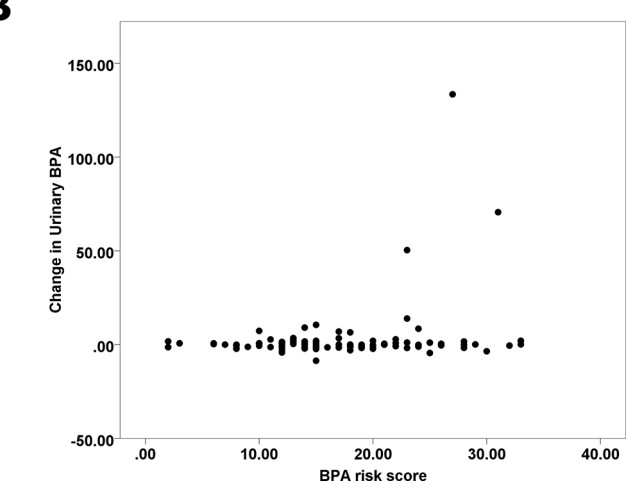

**Figure 1** The effect of a 'real-world' bisphenol A (BPA) avoidance diet on urinary BPA exposure over a 7-day period. (A) Urinary BPA (ng/mL) adjusted for urinary creatinine was plotted at visit 1 before the intervention and at visit 2 after the intervention. Three extreme outliers have been removed. The trajectories of individual participant measurements are shown. (B) Changes in urinary BPA in ng/mL following the intervention diet are plotted against the self-reported BPA risk score.

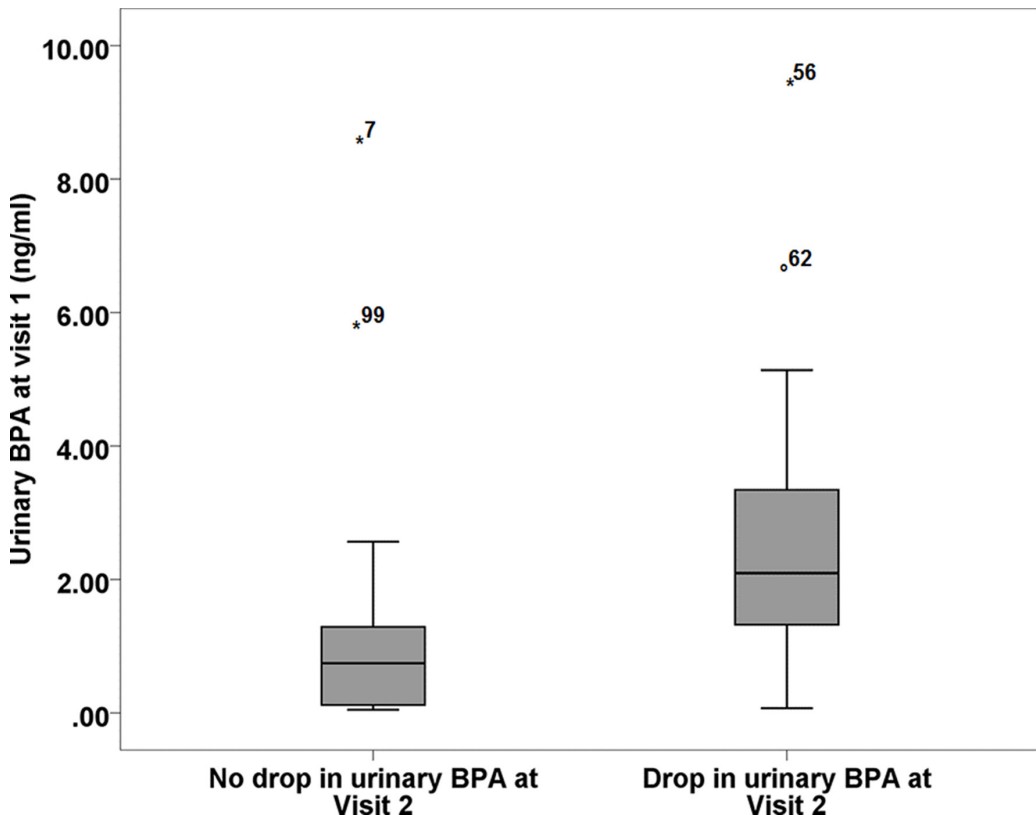

**Figure 2** The effect of baseline urinary bisphenol A (BPA) on the probability of achieving a drop in concentrations following the intervention. This graph illustrates the median urinary BPA adjusted for creatinine at visit 1 prior to the intervention expressed relative to whether or not a reduction in urinary BPA was achieved following the 7-day intervention diet at visit 2. Error bars refer to the IQR of measurement.

participants reported that they felt very restricted. Finally, 66% of participants stated that they would find it hard or very hard to follow the diet long term.

### Qualitative analysis of the effect of following the diet on lifestyle

We identified five over-riding themes in our qualitative analysis of the effect of following the diet on lifestyle. These were (1) the widespread use of plastics possibly containing BPA in food packaging ("almost everything is packaged in plastic"—participant 70, "Literally everything involved plastic"—participant 28). (2) Lack of clarity in labelling of products and packaging potentially containing BPA ("I found it really hard to know what foods I could eat … there is never a guarantee it is BPA free"—participant 43, "The biggest problem was that a lot of packaging doesn't state what type of plastic it is or whether it contains BPA"—participant 74). (3) The perceived restrictions of being on the 'real-world' BPA avoidance diet ("Difficulty eating out, hard to find foods in college or 'out' that hadn't touched BPA. My family had a takeaway on Saturday night and I couldn't eat it"—participant 56, "Sometimes I can't eat/drink what I want because of the recycling number"—participant 112). (4) The impact of eating 'BPA free' was the only positive theme emerging ("I feel I have eaten much more healthily this week … I didn't eat so much junk food"—participant

74, "I ate more vegetables and less chocolate"—participant 83). (5) The impact on shopping habits ("You can't get it all from supermarkets"— Participant 37; "Had to go to more individual food shops"—participant 103).

### DISCUSSION

Exposure to the EDC BPA is ubiquitous,[2] with growing evidence that it may be associated with adverse health outcomes.[4] Here, 94 researcher participants aged 17–19 years designed and undertook a quantitative and qualitative engaged research project designed to assess the potential for reduction of personal exposure to BPA through moderation of diet, which would have utility in a 'real-world' setting. We conclude that the 'real-world' diet designed to reduce BPA exposure had no effect on creatinine-adjusted urinary BPA concentrations in our cohort over a period of 7 days in our dataset.

Although concentrations of urinary BPA in our study cohort were slightly lower at the outset of the study than in others,[18] measurable concentrations were present in the vast majority of our participants. Participants were unable to achieve a reduction in their urinary BPA over the 7-day trial period, despite good compliance to supplied guidelines. Avoidance of BPA was not easily achieved on an individual level in our study population, with qualitative analysis indicating that participants experienced feelings of restriction and difficulties in sourcing BPA-free food

due to inadequate labelling of foods and food packaging. This suggests that the intervention would be difficult to sustain in the longer term.

This work represents the largest group of unrelated participants in a high exposure demographic to date, since previous work has focused on families and related individuals,[16 17] who may share common sources of BPA. Although other population demographics such as young children may have higher concentrations of BPA than our chosen study population,[18] it would not have been possible to do the sort of engaged research project that we envisaged in this group. Our intervention is a 'real-world' diet, designed to a set of guidelines (such as reduction in the usage of tinned foods or foods with high levels of processing), rather than the strict, prescribed diets that have been used in other studies,[16] which suggested that it was possible for participants to reduce their urinary BPA excretion by approximately 60% in a period of just 3 days.[16] In our self-designed, self-administered study this was unachievable. This may reflect the difficulty in identifying and sourcing foods free of BPA in the current commercial environment. Finally, the qualitative thematic analysis has given an indication that adherence to even a 'real-world' BPA reduction diet with fewer restrictions and more choice over the longer term was unlikely in our study population due to difficulties in identifying foodstuffs likely to contain less BPA.

BPA has a terminal half-life of 6 hours.[15] Spot samples may therefore not be as accurate as continuous sampling strategies (24 hours urine collection). However, recent studies suggest that despite its short half-life, measurable BPA remains present for up to 43 hours postfasting, indicating non-food exposures or accumulation in body tissues such as fat.[20] We identified no impact of time of sample collection on BPA concentrations in our sample set, in either creatinine-adjusted or unadjusted data, indicating that our measurements were not influenced by time since the last meal. Spot sampling as used here may therefore represent an acceptable compromise and remains a practical option in the community setting. The large variability in urinary BPA within an individual sampled at different times may also have reduced our ability to observe an effect. This could be facilitated by the use of multiple sampling or pools of multiple urines, but was not feasible within the confines of our study.

Calculating an accurate BPA risk score is challenging. Data were self-reported, and foodstuffs are not labelled for BPA content. It is difficult to generalise across food types and large variations in BPA concentrations occur between different products of the same food type or even different lots of the same product.[1] Foods that were free of BPA-containing packaging (as far as it was possible to tell) may have been highly processed or contain food items from a variety of sources. Highly processed and 'fast' food has previously been demonstrated to be a source of BPA.[21] A study of the temporal trends seen in composite food samples found no change in the overall BPA content of the food, despite large reduction in the BPA content of some individual food items, illustrating the difficulties in effectively excluding BPA from a varied diet.[22] Participants may therefore have changed BPA-containing foods for other, perceived healthier choices, which may still contain BPA by virtue of processing.

BPA enters foodstuffs by leaching from polycarbonate or epoxy resin after manufacture, or by hydrolysis of the polymer itself.[23] The migration rate of BPA increases with higher temperatures,[24] and with time and use, for example, repeated use of polycarbonate water bottles.[25] Exposure to BPA can also occur through routes other than food, including dust ingestion and dermal absorption[26] and this was not taken into account in our study. A study of volunteers who purposefully handled thermal receipts showed an increase in urinary BPA excretion of up to 84%, and their BPA levels took longer to return to pre exposure levels, suggesting a difference in the bioavailability of BPA through skin and oral routes.[27] It is also possible that some manufacturers may have voluntarily reduced the amount of BPA-containing food packaging compared with their previous usage, given the attention that EDCs have received in the media. However, measurable levels of BPA were still detected in the majority of participants in our study, which suggests that there may be other, non-dietary, sources of BPA, and that exposure to BPA remains an issue. We may also have been underpowered to detect subtle changes in urinary BPA, given the heterogeneity in food choice; detection of such effects may need thousands of participants. Finally, our study, like other studies of its type, does not take account of interindividual differences in the metabolism and excretion of BPA arising from differences in genetic background between people. BPA is metabolised primarily by uridine 5′-diphospho-glucuronosyltransferases, and altered activity polymorphisms of these enzymes have been reported.[28]

Emerging evidence suggests that that BPA may be linked to several chronic human health conditions,[6–9 29] suggesting that continued study of the human health effects of BPA exposure is justified. The opinion of the European Food Safety Authority (EFSA), is that while uncertainty over the human health effects of BPA exists, caution should be exercised in ingestion of BPA.[3] Our data suggest that in our study population, it is unlikely that participants could moderate their own BPA exposure in the long term by self-directed modification of diet in a 'real-world' setting, and furthermore, participants would have been reluctant to adopt such a lifestyle change in the longer term due to the restrictions in dietary choice and the effects on day-to-day life. Most of these barriers appear to arise from the pervasiveness of BPA in our food chain, and inadequate labelling of foods packaged in BPA-containing substances. We propose that until a definitive assessment of the health risks of BPA is available, informed choice over whether or not to consume BPA and similar chemicals in foodstuffs should be facilitated by better labelling.

**Author affiliations**
[1]College of Life and Environmental Sciences, University of Exeter, Exeter, UK
[2]Research Projects, St Lukes campus, University of Exeter, Exeter, UK
[3]RNA-Mediated Disease Mechanisms Group, Institute of Biomedical and Clinical Sciences, University of Exeter Medical School, Exeter, UK
[4]National Institute for Health Research Exeter Clinical Research Facility, Royal Devon and Exeter National Health Service Foundation Trust, Exeter, UK
[5]Medical School Building 03.11, University of Exeter Medical School, Exeter, UK
[6]Clyst Vale Community College, Broadclyst, UK
[7]Exeter School, Exeter, UK
[8]South Dartmoor Community College, Ashburton, Devon
[9]Honiton Community College, Honiton, UK
[10]Exeter College, Exeter, UK
[11]Exeter Mathematics School, Exeter, UK

**Correction notice** This article has been corrected since it first published. The location of 'South Dartmoor Comminity College' was corrected to 'Ashburton, Devon'.

**Acknowledgements** We acknowledge the work of the NIHR Exeter Clinical Research Facility in aiding the collection of the urine samples.

**Contributors** TSG contributed to study design and cowrote the paper. NB contributed to study design and participant involvement. BPL managed the technical aspects of the project and reviewed the manuscript. ALK contributed to data entry and interpretation and reviewed the manuscript. BPA SSC members designed and interpreted the study and contributed to the manuscript. MHS carried out the qualitative analysis and reviewed the manuscript. AMS managed sample collection, contributed to study design and reviewed the manuscript. LWH: principal investigator, managed the study, wrote the manuscript.

**Funding** This study was funded by a Wellcome Trust People Award to LWH and TSG (grant no 105162/Z/14/Z). TSG was additionally supported by NERC awards NE/L007010 and NE/N006178/1

**Competing interests** None declared.

**Patient consent** Obtained.

**Ethics approval** Ethical permission was granted by the University of Exeter Medical School Ethics Committee (reference no 15/07/074), and the study was carried out in accordance with the Declaration of Helsinki.

**Provenance and peer review** Not commissioned; externally peer reviewed.

**Data sharing statement** Data are available on reasonable request by emailing the corresponding author (L.W.Harries@exeter.ac.uk).

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
