## [Reviewer comments · BMJ Open]

ARTICLE DETAILS

TITLE (PROVISIONAL)	An engaged research study to assess the effect of a 'real-world' dietary intervention on urinary bisphenol A (BPA) levels in teenagers
AUTHORS	Galloway, Tamara; Baglin, Nigel; Lee, Benjamin; Kocur, Anna; Shepherd, Maggie; Steele, Anna; BPA Schools, Study Consortium; Harries, Lorna

VERSION 1 – REVIEW

REVIEWER	Maribel Casas Barcelona Institute for Global Health (ISGlobal). Spain.
REVIEW RETURNED	10-Aug-2017

GENERAL COMMENTS	This study represents an intervention diet in 98 adolescents aged 17-19 years from Exeter, UK. Urinary BPA concentrations were measured in a spot urine sample before and after 7 days of intervention. The dietary intervention was designed by students and they also performed a qualitative assessment on whether it would be possible to follow the same diet after the intervention. A BPA risk score was constructed for each child based on information on packaging and food/drink ingested of each child. The authors did not show any difference in BPA concentrations before and after the intervention. The students reported the difficulty in identifying those foods that could contain BPA. Hereby I include some comments: - In authors list I would include “BPA schools study consortium#” at the end.- This sentence in the abstract is not clear for me: “Questionnaires and freeform comments on the ease of use were collected for qualitative analyses”.- Specify the place where the study is developed.- A part of reference 6 when referring to epidemiological studies the authors could include some of the recent reviews done on the potential health effects linked with BPA exposure such as the one of Braun 2017 (Nat Rev Endocrinol. 2017 Mar;13(3):161-173) on obesity and neurodevelopment and many other ones.- In the abstract the age range is 18-19 but in the methods is 17-19; please correct it.- Describe how many adolescents were located in the same school.- This sentence is not completely clear: “Risk scores from the final day of the intervention only were also taken, since this is most relevant to the sample collected at visit 2.”- It would be nice to describe how many students were initially invited and how many finally participated in the intervention; by
--

	including a flowchart for example.  - “Each participant was sampled in the same time-slot at both visits...” – please, specify whether these both visits are before and after the intervention or during it, it is not clear enough. Is visit 1 before and visit 2 after the intervention? - BPA concentrations taken at different time points during a week present very low reproducibility (<0.50 intraclass correlation coefficient). Therefore, the authors might consider that the difference observed between BPA concentrations before and after the intervention may just reflect BPA own variability. A spot sample only reflects BPA exposure in the last 6 hours (as pointed out in the introduction). The collection of multiple urines before and after the intervention (and analyze pools of multiple urines for example) could have facilitated the observation of changes in BPA levels. - Specify the LOQ of BPA in the methods sections not in results. - This sentence is not clear: “Study population demographics for urinary BPA adjusted for creatinine at visits 1 and 2 were 173 assessed to generate a BPA continuous variable.” - Concentrations of BPA in this population are not very high but the authors stated at the beginning that this is one of the populations with a higher exposure. I am wondering why the authors did not initially perform a cross-sectional study in different population settings to know the levels of BPA in their population (children, adolescents, and adults) and then select the group with the higher exposure. Young children for example have high BPA levels but I can understand that the intervention in this group would be much difficult to perform. This can be discussed in the discussion. - For me it is not clear the construction of the BPA risk score (from 0 to XX?) – It would be nice to show an example in the supplementary material. - I wonder whether school can have an effect on the intervention. How was school treated in the statistical analysis? - Please, if the final number is 98 but not 99 or 104 authors should always refer to 98. - Define what “exposure to estrogens” means. - This sentence must be included in the discussion not in the results section: “We conclude that the ‘real world’ diet designed to reduce BPA exposure had no effect on creatinine-adjusted urinary BPA concentrations in our cohort over a period of 7 days in our dataset.” - “...their shopping took longer” – i do not know if adolescents between 17 and 19 years old make their own shopping...
--	---

REVIEWER	Sheela Sathyanarayana University of Washington Seattle Children's Research Institute USA
REVIEW RETURNED	17-Aug-2017

GENERAL COMMENTS	Summary: This study assesses the ability of a “real life” intervention to reduce BPA exposures. BPA exposure is associated human health impacts in numerous animal and human studies, making it an important study to conduct. Few studies have examined how and if exposures to endocrine disrupting chemicals can be reduced through individual behavioral changes. Some minor issues should be addressed before publication. Minor Comments:  1. While the authors state that sources of exposure may be other than foods, it would be good to talk about how concentrations have declined over time and what risk assessments show about the proportion of exposure coming from foods. The pre-intervention concentrations at baseline are approximately half of those in Rudel et al. It may be that manufacturers that changed their practices in that time so that food is truly not a major source in the general population. 2. The “real life” scenario needs to be better described for the students. What did they actually ask the students to do besides read the information? 3. Why was there only one urinary measurement before, during, after? Given the inter-individual variability, this makes it hard to know if exposure misclassification led to the results observed. 4. When were samples collected before and after the intervention? 5. If they learned about the study in the class, could have they changed their practices before the intervention period? 6. The authors need to better describe how they assessed adherence to the intervention.
--

VERSION 1 – AUTHOR RESPONSE

Reviewer: 1

1) In authors list I would include “BPA schools study consortium#” at the end.

Response: It is our understanding that the last position on the author list is the senior author, who has responsibility for the study; it is therefore not appropriate to place the schools consortium at this position. We have moved the consortium to second to last, in accordance with the reviewer’s wishes.

2) This sentence in the abstract is not clear for me: “Questionnaires and freeform comments on the ease of use were collected for qualitative analyses”.

Response: We have omitted this sentence for clarity, as it is also mentioned as an outcome measure.

2) Specify the place where the study is developed.

Response: We have included information on where the study was designed (page 7, para 3, line 141)

3) A part of reference 6 when referring to epidemiological studies the authors could include some of the recent reviews done on the potential health effects linked with BPA exposure such as the one of

Braun 2017 (Nat Rev Endocrinol. 2017 Mar;13(3):161-173) on obesity and neurodevelopment and many other ones.

Response: We have added some references on human health outcomes including the suggested reference (page 5, para 1, line 82)

4) In the abstract the age range is 18-19 but in the methods is 17-19; please correct it.

Response: The correct age range is 17- 19 years, we have corrected the error in the abstract.

5) Describe how many adolescents were located in the same school.

Response: This information is given in Supplementary table 1.

6) This sentence is not completely clear: "Risk scores from the final day of the intervention only were also taken, since this is most relevant to the sample collected at visit 2."

Response: This sentence refers to the possibility that the short half-life of BPA means that the levels in urine probably reflect the most recent exposure. We have amended the sentence to make it clearer (page 8, para 8, line 152).

7) It would be nice to describe how many students were initially invited and how many finally participated in the intervention; by including a flowchart for example.

Response: We have clarified the numbers of students invited, the number of students participating and the number on whom a complete dataset was produced in the manuscript (page 6, para 3, line 121 and page 10, para 2, line 213).

8) "Each participant was sampled in the same time-slot at both visits..." – please, specify whether these both visits are before and after the intervention or during it, it is not clear enough. Is visit 1 before and visit 2 after the intervention?

Response: Each participant was sampled twice, once just before and once just after the intervention. We had to stagger the collections due to the large number of samples that needed to be taken and processed, but each participant was sampled at the same timeslot at visit one and visit 2 to account for circadian variation in BPA levels. We have amended the manuscript to make this clearer (Page 8, para 2, line 164).

9) BPA concentrations taken at different time points during a week present very low reproducibility (<0.50 intraclass correlation coefficient). Therefore, the authors might consider that the difference observed between BPA concentrations before and after the intervention may just reflect BPA own variability. A spot sample only reflects BPA exposure in the last 6 hours (as pointed out in the introduction). The collection of multiple urines before and after the intervention (and analyze pools of multiple urines for example) could have facilitated the observation of changes in BPA levels.

Response: Yes, we appreciate that intra-individual variation in BPA levels may complicate analysis and reduce the chances of observing an effect, and we do discuss the issues around the use of spot samples in the discussion. Although we would have liked to take multiple samples, or use pooled urines this was not feasible due to the large number of participants needing to be simultaneously sampled, and the constraints on the students' timetables just before their final exams. We acknowledge the issue however, and have added a comment to reflect this in the discussion (Page 16, para 2, line 328).

10) Specify the LOQ of BPA in the methods sections not in results.

Response: We have amended the manuscript as requested (page 9, para 1, line 180).

11) This sentence is not clear: "Study population demographics for urinary BPA adjusted for creatinine at visits 1 and 2 were assessed to generate a BPA continuous variable."

Response: We have clarified the wording of this section (Page 9, para 2, line 185).

12) Concentrations of BPA in this population are not very high but the authors stated at the beginning that this is one of the populations with a higher exposure. I am wondering why the authors did not initially perform a cross-sectional study in different population settings to know the levels of BPA in their population (children, adolescents, and adults) and then select the group with the higher exposure. Young children for example have high BPA levels but I can understand that the intervention in this group would be much difficult to perform. This can be discussed in the discussion.

Response: The choice of study population was firstly led by the observation from NHANES that teenagers had one of the highest reported BPA exposures (Calafat EHP 2008); this study had effectively already done the cross sectional analysis suggested by the reviewer. Whilst younger children may have had higher levels of BPA than the levels seen in teenagers, it would not have been feasible to undertake a study of the sort we envisaged with them. We have added a note to this effect in the discussion of our manuscript (page 15, para 3, line 307).

13) For me it is not clear the construction of the BPA risk score (from 0 to XX?) – It would be nice to show an example in the supplementary material.

Response: Each individual incidence of consumption of food or drinks containing or packaged in BPA was given a score of 1. At the end of the intervention, these values were totalled to give the overall BPA risk score. A fuller explanation of this is now presented in the methods (page 8, para 1 line 149). The mean, median and 95% Confidence Intervals for risk score are given in table 1.

14) I wonder whether school can have an effect on the intervention. How was school treated in the statistical analysis?

Response: School had no effect on the intervention. We tried our analysis with and without, but did not in the end include it as a confounder. We have included a note to this effect in the manuscript (page 12, para 2, line 237).

15) Please, if the final number is 98 but not 99 or 104 authors should always refer to 98.

Response: This is a difficult point, as some students participated in the intervention but were unable to provide a sample at one or other visit but still participated fully in all other aspects of the study. For this reason, we are reluctant to omit them from the sample set. Instead, we have made clear how many students provided a full dataset in the manuscript (Page 10, para 2, line 213).

16) Define what "exposure to estrogens" means.

Response: This refers mainly to contraceptive usage, but we were not able to refer to it directly as such following directive from our ethics review board.

17) This sentence must be included in the discussion not in the results section: “We conclude that the ‘real world’ diet designed to reduce BPA exposure had no effect on creatinine-adjusted urinary BPA concentrations in our cohort over a period of 7 days in our dataset.”

Response: We have moved the sentence to the discussion as requested (Page 14, para 2, line 291).

18) “...their shopping took longer” – i do not know if adolescents between 17 and 19 years old make their own shopping...

Response: In actual fact, the students who participated in our study took responsibility for the shopping during the week of the intervention – many of their families also followed the diet!

Reviewer: 2

1. While the authors state that sources of exposure may be other than foods, it would be good to talk about how concentrations have declined over time and what risk assessments show about the proportion of exposure coming from foods. The pre-intervention concentrations at baseline are approximately half of those in Rudel et al. It may be that manufacturers that changed their practices in that time so that food is truly not a major source in the general population.

Response: This is certainly a possibility given the interest surrounding BPA in the media. We have added a note to this effect to the discussion (page 17, para 2, line 352).

2. The “real life” scenario needs to be better described for the students. What did they actually ask the students to do besides read the information?

Response: A set of dietary guidelines was co-designed with the students based on known sources of BPA. We asked students to use these guidelines to avoid BPA wherever possible, and to note down potential incidences of exposure in their diet diaries. We also asked them to note down the type and amount of packaging. We have added more information on this to the methods section of our manuscript (Page 7, para 4, line 145).

3. Why was there only one urinary measurement before, during, after? Given the inter-individual variability, this makes it hard to know if exposure misclassification led to the results observed.

Response: We agree that multiple samples would have given us a better measure, but this was not feasible due to the large number of participants needing to be simultaneously sampled, and the constraints on the students’ timetables just before their final exams. We acknowledge the issue however, and have added a comment to reflect this in the discussion (Page 16, para 2, line 328).

4. When were samples collected before and after the intervention?

Response: At visit 1, samples were collected during the morning just prior to the students commencing the intervention at lunchtime. At visit 2, samples were collected during the morning just prior to students recommencing their normal diet for lunch. We have added this information to the manuscript (Page 8, para 2, line 167).

5. If they learned about the study in the class, could have they changed their practices before the intervention period?

Response: This is an area that we were also concerned about, so students were asked specifically not to make any dietary changes before the intervention. We have added a note to this effect in the manuscript (Page 8, para 1, line 156).

6. The authors need to better describe how they assessed adherence to the intervention.

Response: Adherence was measured using our BPA risk score (see Page 8, para 1, line 149). We did not experience reduced adherence as an issue, probably because participants were fully engaged in the accuracy of the study in their role as student researchers. In our experience, although students did occasionally consume items known to contain BPA, these incidences were noted down in their diet diaries and accounted for.

VERSION 2 – REVIEW

REVIEWER	Maribel Casas Barcelona Institute for Global Health (ISGlobal)
REVIEW RETURNED	06-Oct-2017

GENERAL COMMENTS	-IMPORTANT: Please, remove the names of the participants in Supplementary information file 1. Data should be always anonymized. -Do not start the sentence with a number; write “a total of” before it or write the number. -I do not agree on that comment regarding the total number of participants: “This is a difficult point, as some students participated in the intervention but were unable to provide a sample at one or other visit but still participated fully in all other aspects of the study. For this reason, we are reluctant to omit them from the sample set. Instead, we have made clear how many students provided a full dataset in the manuscript (Page 10, para 2, line 213).” The final dataset should include those children with all the information (diet and particularly urines). -This sentence should be revised: “Adherence was assessed using a ‘BPA risk score’ based on instances of known or suspected exposure for each participant was then calculated, whereby each individual incidence of potential BPA exposure was given a score”. -This sentence is very difficult to understand: “Risk scores from the final day of the intervention only were also considered, since the half-life of BPA means that this is most relevant to the sample collected at visit 2”. -Please, include the number of students in each school in the main text as well as the supplementary table. -In participating group section I would say that adolescents were chosen because it has been shown that they have higher concentrations than adults (and include the reference of the NHANES study). In the NHANES study children have higher concentrations than adolescents and only saying “high” it seems that adolescents are those with the highest exposure. -The authors explained better the construction of the BPA risk score than in the first version of the manuscript; however, it would be nice to include an example in the supplementary material.
--

VERSION 2 – AUTHOR RESPONSE

We thank the reviewer and the editor for their continued support of our manuscript. Please find our responses to the issues raised here.

1) IMPORTANT: Please, remove the names of the participants in Supplementary information file 1. Data should be always anonymized.

Response: We have removed supplementary table 1, but added the school affiliations to the author list. However, it should be noted that the individuals noted in this table are authors, as well as participants as they were instrumental in the design, implementation and analysis of the study. This is an engaged research project, and removing the students from the author list negates the whole enterprise. There is no way to link individual participants with individual data in the results, so we feel strongly that their names should be included, as one would with any other author. Accordingly, we have included a file with the names of consortium members that can be included if the editor feels, as we do, that it is appropriate.

2) Do not start the sentence with a number; write “a total of” before it or write the number.

Response: We have reworded the manuscript throughout accordingly.

3) I do not agree on that comment regarding the total number of participants: “This is a difficult point, as some students participated in the intervention but were unable to provide a sample at one or other visit but still participated fully in all other aspects of the study. For this reason, we are reluctant to omit them from the sample set. Instead, we have made clear how many students provided a full dataset in the manuscript (Page 10, para 2, line 213).” The final dataset should include those children with all the information (diet and particularly urines).

Response: We should clarify that the analyses were carried out only with the final sample set of 94 participants. We have amended the manuscript to clarify this and refer throughout to a dataset of 94 participants, except on the first mention when we document the total number invited to participate (line 126)

4) This sentence should be revised: “Adherence was assessed using a ‘BPA risk score’ based on instances of known or suspected exposure for each participant was then calculated, whereby each individual incidence of potential BPA exposure was given a score”.

Response: We have reworded the sentence as requested (line 158).

5) This sentence is very difficult to understand: “Risk scores from the final day of the intervention only were also considered, since the half-life of BPA means that this is most relevant to the sample collected at visit 2”.

Response: We have reworded the sentence as requested (line 162)

6) Please, include the number of students in each school in the main text as well as the supplementary table.

Response: We have added this information to the manuscript (line 128)

7) In participating group section I would say that adolescents were chosen because it has been shown that they have higher concentrations than adults (and include the reference of the NHANES

study). In the NHANES study children have higher concentrations than adolescents and only saying “high” it seems that adolescents are those with the highest exposure.

Response: We have amended the manuscript accordingly (line 125).

8) The authors explained better the construction of the BPA risk score than in the first version of the manuscript; however, it would be nice to include an example in the supplementary material.

Response: An example has been included as requested (new supplementary information file 3).

VERSION 3 – REVIEW

REVIEWER	Maribel Casas Barcelona Institute for Global Health (ISGlobal)
REVIEW RETURNED	22-Nov-2017

GENERAL COMMENTS	Authors have replied to all my comments properly and I appreciate that they pointed out that the analysis was done in only the 94 participants with the complete dataset. In relation to that it would be nice that Table 1 shows the descriptive statistics for the 94 participants included and not for 98 or 99.
---

VERSION 3 – AUTHOR RESPONSE

We have made the requested amendment. Table 1 now contains only data from individuals in the complete dataset of 94.